# Monocyte Chemoattractant Protein-1 (MCP-1), Activin-A and Clusterin in Children and Adolescents with Obesity or Type-1 Diabetes Mellitus

**DOI:** 10.3390/diagnostics14040450

**Published:** 2024-02-19

**Authors:** Eirini Kostopoulou, Dimitra Kalavrizioti, Panagiota Davoulou, Evangelos Papachristou, Xenophon Sinopidis, Sotirios Fouzas, Theodore Dassios, Despoina Gkentzi, Stavroula Ioanna Kyriakou, Ageliki Karatza, Gabriel Dimitriou, Dimitrios Goumenos, Bessie E. Spiliotis, Panagiotis Plotas, Marios Papasotiriou

**Affiliations:** 1Division of Pediatric Endocrinology, Department of Pediatrics, University Hospital of Patras, School of Medicine, University of Patras, 26504 Patras, Greece; eirinikost@gmail.com (E.K.); besspil@gmail.com (B.E.S.); 2Department of Nephrology and Kidney Transplantation, University Hospital of Patras, School of Medicine, University of Patras, 26504 Patras, Greece; dkalavrizioti@yahoo.com (D.K.); pdavoulou@gmail.com (P.D.); epapachr@upatras.gr (E.P.); dgoumenos@upatras.gr (D.G.); mpapasotir@upatras.gr (M.P.); 3Department of Pediatric Surgery, University Hospital of Patras, School of Medicine, University of Patras, 26504 Patras, Greece; stavriannakyr@gmail.com; 4Department of Pediatrics, University Hospital of Patras, School of Medicine, University of Patras, 26504 Patras, Greece; sfouzas@upatras.gr (S.F.); tdassios@upatras.gr (T.D.); gkentzid@upatras.gr (D.G.); karatza@upatras.gr (A.K.); gdim@upatras.gr (G.D.); 5Department of Speech and Language Therapy, School of Health Rehabilitation Sciences, University of Patras, 26504 Patras, Greece; pplotas@upatras.gr

**Keywords:** monocyte chemoattractant protein-1 (MCP-1), activin-A, clusterin, type-1 diabetes mellitus (T1DM), obesity, children, adolescents, diagnosis, prognosis

## Abstract

Inflammation plays a crucial role in diabetes and obesity through macrophage activation. Macrophage chemoattractant protein-1 (MCP-1), activin-A, and clusterin are chemokines with known roles in diabetes and obesity. The aim of this study is to investigate their possible diagnostic and/or early prognostic values in children and adolescents with obesity and type-1 diabetes mellitus (T1DM). Methods: We obtained serum samples from children and adolescents with a history of T1DM or obesity, in order to measure and compare MCP-1, activin-A, and clusterin concentrations. Results: Forty-three subjects were included in each of the three groups (controls, T1DM, and obesity). MCP-1 values were positively correlated to BMI z-score. Activin-A was increased in children with obesity compared to the control group. A trend for higher values was detected in children with T1DM. MCP-1 and activin-A levels were positively correlated. Clusterin levels showed a trend towards lower values in children with T1DM or obesity compared to the control group and were negatively correlated to renal function. Conclusions: The inflammation markers MCP-1, activin-A, and clusterin are not altered in children with T1DM. Conversely, obesity in children is positively correlated to serum MCP-1 values and characterized by higher activin-A levels, which may reflect an already established systematic inflammation with obesity since childhood.

## 1. Introduction

Pediatric obesity and type-1 diabetes mellitus (T1DM) represent two of the most common chronic diseases in childhood, with a rising prevalence worldwide [1]. Both diseases increase the risk of short- and long-term complications impairing the physical and psychological state, as well as quality and duration of life [1]. Progress in the understanding of the pathophysiology of these diseases has revealed that both are characterized by chronic inflammation and immune dysregulation. In obesity, macrophage accumulation in the adipose tissue plays a significant role in the development of low-grade inflammation through impaired cytokine secretion. This pro-inflammatory activity, which seems to emanate partly from the adipose tissue, is a common feature of many obesity-associated comorbidities in children and adults [2,3,4]. As opposed to obesity, which is characterized by systemic inflammation, T1DM is characterized by the infiltration of the pancreas by immune cells activating a selective, specific inflammatory process that affects the insulin-producing pancreatic β-cells [5]. In this autoimmune setting, regulatory T cells, CD4+ and CD8+ T cells, and macrophages have been shown to mediate the islet inflammation [6].

The quantification of serum or urine molecules that may be involved in the pathophysiology of these two diseases and their complications has recently gained attention. Such molecules include monocyte chemoattractant protein-1 (MCP-1), activin-A, and clusterin.

MCP-1 was the first discovered, and widely studied, human chemokine. It mobilizes and stimulates leukocytes, particularly monocytes and macrophages, and is also involved in the recruitment of memory T cells [7]. These cells produce inflammatory cytokines such as IL-1 and IL-6, as well as superoxide, further contributing to the inflammatory process.

The gene expression of MCP-1 has been found to be increased in the visceral and subcutaneous adipose tissues of patients with obesity compared to lean control group patients [8]. The plasma concentrations of MCP-1 have also been found to be increased in adults and children with obesity [9,10,11], and decrease with weight loss [12]. Serum MCP-1 has been found to be significantly elevated in patients with type-2 diabetes (T2DM). It has been proposed that in patients with T1DM and T2DM, particularly those with concomitant diabetic nephropathy and diabetic retinopathy, oxidative stress increases locally produced MCP-1, which triggers macrophage-induced inflammation [13,14,15].

In addition, activins are members of the transforming growth factor (TGF)-β family. They regulate the apoptosis, proliferation, and differentiation of a variety of cells [16]. Activin-A, a homodimer of bA subunits, is expressed in adipocyte progenitors and in mature adipocytes, and plays a key role in the proliferation of human adipocyte progenitors, but also in the inhibition of their differentiation [17]. The inhibition of activin-A enhances adipocyte differentiation in an autocrine/paracrine manner. Additionally, activin-A is a mediator of inflammation and is also involved in glucose regulation, as it is increased in adults with impaired glucose tolerance (IGT) or impaired fasting glucose (IFG) [18].

Furthermore, clusterin is a molecular chaperone that participates in the folding of proteins [19]. It is produced by many human tissues and is highly expressed in the adipocytes of patients with obesity compared to lean subjects, while it also acts as a sensor of oxidative stress. Plasma clusterin has been associated with diabetes mellitus, while polymorphisms of the *CLU* gene, the gene encoding clusterin, are linked to impaired insulin secretion and insulin resistance [20].

The aim of the present study is to investigate the possible association of serum concentrations of MCP-1, activin A, and clusterin with obesity or T1DM in children and adolescents.

## 2. Materials and Methods

### 2.1. Study Population

Children and adolescents with T1DM or obesity who were between the ages of 2–18, and who were routinely followed-up with in the Outpatient Clinic of Pediatric Endocrinology of the University Hospital of Patras, a tertiary hospital, were included in this study. Children who were newly diagnosed T1DM (less than six months) were excluded. Moreover, patients or controls with a history of acute febrile illness during the previous 2 weeks, other endocrinopathies (hypothyroidism, adrenal insufficiency, etc.), or chronic diseases (cancer, asthma, or malabsorption) were excluded from this study. As controls, we used children and adolescents who were externally referred to the outpatient clinic and subsequently proved to have no substantial medical issues. The participants’ weight and height were measured in the morning and BMI was calculated as weight divided by height squared (kg/m^2^). Obesity was defined as a BMI z-score > 2 [21]. The pubertal status of the patients was also estimated using Tanner staging.

This study was approved by the Research Ethics Committee of the University Hospital of Patras (number of approval: 353/02/09/2015) and was conducted in accordance with the Helsinki Declaration, as revised in 2013. Written informed consent was obtained from the parents or legal representatives of the participating children and adolescents, and informed assent was obtained from the participants. All samples were obtained between January 2021 and December 2022, and were measured between May and July 2023.

### 2.2. Serum MCP-1, Activin-A, Clusterin, and Standard Biochemical Measurements

All serum samples obtained from the control group participants and the patients were appropriately stored at −80 °C until assayed after separation from clotted blood by centrifugation for 10 min at 1200× *g* in 4 °C. MCP-1, activin-A, and clusterin concentrations were measured using commercially available sandwich ELISA kits (MCP-1: DCP00; activin-A: DAC00B; clusterin: DCLU00; R&D Systems, Minneapolis, MN, USA). The tests were performed according to the manufacturer’s recommended protocols.

The participants’ lipid profiles (total serum cholesterol, high-density lipoprotein (HDL), low-density lipoprotein (LDL), triglyceride, creatinine levels, and hemoglobin A1c (HbA1c)) were also determined using an automated analyzer (ADVIA^®^ 2400 Chemistry System, Siemens, Athens, Greece). Epidemiological and clinical data including the age, duration of diabetes, glucose control expressed as HbA1c concentrations, and severity of obesity, were documented using the patients’ medical records. Estimated glomerular filtration rate (eGFR) was calculated using the revised bedside Schwartz formula, as indicated by the kidney disease improving global outcomes (KDIGO) guidelines [22].

### 2.3. Statistical Analysis

Quantitative variables were presented by the mean and standard deviation when normally distributed, or median and interquartile range [IQR] when skewed. The Kolmogorov Smirnov test was used for normality analysis. Categorical data were presented as frequencies and percentages (*n*, %). In order to test the mean differences of MCP-1, activin-A, and clusterin serum levels between the control participants and children with T1DM or obesity, we used ANOVA or Kruskal–Wallis tests with Bonferroni or Dunn’s post-hoc analyses for normally distributed data and skewed data, respectively. Furthermore, patients with T1DM were divided into two groups according to their mean glycated hemoglobin (HbA1c) levels, calculated using all measurements during the last year of their follow up, (i.e., Group 1: patients with a mean HbA1c <7%, and Group 2: patients with a mean HbA1c >7%). Comparisons between HbA1c and Tanner groups, and measured markers were performed using the independent samples t-tests and Mann–Whitney tests in case of a violation of normality. Moreover, correlation Spearman’s rho was used to test the relationship between demographic and clinical or biochemical variables with investigated serum markers. All analyses were carried out using the SPSS statistical package (version 16.0 SPSS Inc., Chicago, IL, USA) and GraphPad Prism (version 5.00 for Windows, GraphPad Software, San Diego, CA, USA). All statistical tests were two-sided, and significance was set at *p* < 0.05.

## 3. Results

Overall, 129 children and adolescents were included in this study, and their baseline demographic characteristics are shown in Table 1. Each group (control group, patients with T1DM, and patients with obesity) included 43 children. The mean duration of T1DM was 3.61 ± 2.72 years. There were thirty-one children (72%) with a disease duration of more than one year and nine children (20%) with a duration of more than five years. Well controlled T1DM (defined as HbA1c < 7%) was observed in 19 children (44.2%). Mean HbA1c levels were 7.37 ± 1.06%. One third (33.3%) of the patients were pre-pubertal. All patients had normal eGFR and creatinine levels. The biochemical characteristics of the participating children and adolescents are presented in Table 1 as well.

### 3.1. MCP-1 Outcomes

Comparisons of MCP-1 serum levels did not show any significant differences between control participants and patients with obesity or T1DM (Table 2 and Figure 1A). Post-hoc analyses did not reveal any difference between patients with obesity and T1DM. Moreover, patients with different durations of T1DM, either more than 1 year or more than 5 years, did not show significantly different serum MCP-1 levels. Better glucose control with a mean HbA1c of less than 7% did not affect MCP-1 levels. Finally, the pubertal status, as expressed with Tanner stages, of patients with obesity did not affect serum MCP-1 levels.

In patients with T1DM, MCP-1 serum levels showed a positive correlation with BMI z-score (Spearman rho 0.357, *p* = 0.045) and an inverse correlation to TSH levels (Spearman rho −0.32, *p* = 0.044). No significant correlation was found in patients with obesity. When all study participants were examined together, serum MCP-1 levels correlated positively to BMI z-score (Spearman rho 0.331, *p* = 0.025). No significant correlations were found between total cholesterol, LDL, HDL, or triglycerides, and MCP-1 serum levels when examined either in the total population of the cohort or separately in patients with T1DM or obesity.

### 3.2. Activin-A Outcomes

Serum activin-A levels were overall found to be significantly increased in patients with obesity and T1DM in comparison to control group participants (Table 2 and Figure 1B). In post-hoc analyses, only patients with obesity had significantly increased serum MCP-1 levels in comparison to control participants (*p* = 0.014). Moreover, patients with established T1DM for five or more years (*N* = 9) had significantly increased serum activin-A levels in comparison to patients whose T1DM had been established for less than five years (*p* = 0.007). Better glucose control with mean HbA1c of less than 7% did not affect activin-A levels. Finally, in patients with obesity, pubertal status, as expressed by Tanner stages, did not affect serum activin-A levels.

When all study participants were examined together, serum activin-A levels did not correlate to any biochemical or other indices. In patients with T1DM, serum activin-A levels did not show any correlation with examined indices separately. Nevertheless, when correlations were examined in patients with obesity, serum activin-A levels were negatively correlated to LDL (Spearman rho −0.522, *p* = 0.008) and FT4 values (Spearman rho −0.399, *p* = 0.032).

### 3.3. Clusterin Outcomes

Serum clusterin levels did not show significant differences between the control participants and patients with either obesity or T1DM, although in both groups they were slightly and comparably decreased (Table 2 and Figure 1C). Moreover, patients with a different duration of T1DM (either more than 1 year or more than 5 years) did not show significantly different serum clusterin levels. Better glucose control with mean HbA1c levels of less than 7% did not affect clusterin levels. Finally, in patients with obesity, different pubertal status, as expressed with Tanner stages, did not affect serum clusterin levels.

Serum clusterin levels did not show any significant correlations to either serum MCP-1, activin-A, or other clinical or biochemical indices in patients with T1DM or obesity. Nevertheless, when correlations were examined in the control group and all patients combined, serum clusterin levels demonstrated a positive correlation with age (Spearman rho 0.268, *p* = 0.045) and total cholesterol (Spearman rho 0.312, *p* = 0.039), and a negative correlation to kidney function, as expressed with eGFR (Spearman rho −0.391, *p* = 0.025).

## 4. Discussion

One of the most important findings of the present study is the significantly elevated activin-A levels in children and adolescents with obesity, but not in those with T1DM. This may suggest that activin-A has a more important role in adipose tissue expansion than in glucose regulation. Alternatively, the inflammation process may be more activated in obesity than in T1DM, at least during childhood and adolescence.

### 4.1. MCP-1 and Obesity

MCP-1 signaling has been directly associated with the development of obesity. In adults with obesity, MCP-1 seems to be produced as a response resulting from the exposure of adipocytes to inflammatory cytokines and fatty acids, further propagating the inflammatory response. MCP-1 is also produced by other cells, including hepatocytes, skeletal muscle cells, monocytes, vascular smooth muscle, and endothelial cells [23].

It is believed that MCP-1 triggers the recruitment of monocytes that convert to mature adipose tissue macrophages in inflamed adipose and vascular tissue, contributing to systemic inflammation [24] and also playing a role in obesity-related complications [11,25,26]. Inflammation of the adipose tissue is mediated by inflammatory cytokines, such as IL1-1β, IL-6, IL-8, IL-10, IL-12, and Tumor Necrosis Factor (TNF-α), as well as chemokines including MCP-1, which are produced by M1 macrophages in the adipose tissue of individuals with obesity [11,27,28,29]. In addition, serum MCP-1 has been associated with serum high-sensitive C-reactive protein (hsCRP), plasma fibrinogen, and combined carotid artery intimal-media thickness (cIMT), all of which represent risk factors for cardiovascular disease [14,30,31]. In the pediatric population, increased waist-to-hip ratio has been associated with elevated markers of inflammation, including CRP, IL-6, and MCP-1, as well as with markers of endothelial dysfunction, such as soluble intracellular cell adhesion molecule-1 (sICAM) [32]. In addition, overweightness and obesity in children have been associated with elevated IL-6, CRP, and MCP-1, and CRP has been positively correlated with MCP-1, IL-6, and IL-10 [33]. Furthermore, children with MCP-1 promoter polymorphism have a significantly higher tendency to have thicker cIMT compared to children without the polymorphism [34].

Protein and mRNA MCP-1 levels are higher in atherosclerotic lesions [35]. Similarly, MCP-1 has been correlated with complications related to atherosclerosis, such as ischemic stroke [36], myocardial infarction, and cardiovascular disease mortality [14]. The above correlations are shown to be stronger in adults with obesity compared to normal-weight individuals [14,37,38].

The findings of the present study only partially support the reported findings from adult populations, since we only observed a positive correlation between MCP-1 and BMI z-score, and not significant differences in serum MCP-1 concentrations between children and adolescents with obesity and the control participants. Hypothetically, this may be explained by the absence of obesity-related complications in our population, and likely a more favorable inflammation state due to a younger age.

Data from the literature regarding MCP-1 concentrations in children with obesity are limited. There is only one study in the pediatric population, which showed increased MCP-1 concentrations in Mexican children with obesity and dyslipidemia [9]. It should be noted, however, that children of Mexican descent are considered high-risk for obesity-related complications, and the children included in this study had dyslipidemia, whereas our population had no comorbidities such as dyslipidemia, hypertension, or insulin resistance.

### 4.2. MCP-1 and Type-2 Diabetes

According to the literature, MCP-1 is elevated in adult patients with T2DM, particularly those with diabetic complications such as diabetic nephropathy [39]. The proposed underlying mechanism involves the recruitment and activation of monocytes/macrophages by locally produced MCP-1 [40]. Urinary MCP-1 levels have also been found to be elevated in patients with diabetic nephropathy compared to healthy control groups [41,42]. In the same context, MCP-1 has also been proposed as a prognostic marker for the progression of diabetic nephropathy in patients with macroalbuminuria and eGFR decline, regardless of the severity of the albuminuria [43,44].

### 4.3. MCP-1 and Type-1 Diabetes

With regard to the association between serum MCP-1 levels and T1DM, MCP-1 mRNA expression has been found increased in diabetic mice during the early phases of insulitis, as well as at later stages of diabetes, suggesting the possible involvement of β-cell-derived MCP-1 in the recruitment of mononuclear cells into pancreatic islets [45,46]. Insulitis, an inflammatory process mediated by cytotoxic T-cell destruction of the insulin-producing β cells, is the main histopathological finding in patients with T1DM [47]. Furthermore, a study by Chiarelli et al. demonstrated that in patients with T1DM and early renal complications, MCP-1 biosynthesis was induced via enhanced oxidative stress [48].

In addition, several studies support the causative role of serum MCP-1 in diabetic retinopathy both in patients with T1DM and T2DM [49,50]. Urinary MCP-1 has also been found to be elevated in patients with proliferative diabetic retinopathy, and has been shown to have a predictive role for the advent of diabetic retinopathy [44].

To our knowledge, the present study is the first to investigate the possible involvement of MCP-1 in the pathogenesis of T1DM in children and adolescents. Our data showed no significant differences in serum MCP-1 levels between children and adolescents with T1DM and the control participants. Due to the lack of data from other studies, further research in larger populations is required so that the possible role of MCP-1 in T1DM in children and adolescents can be elucidated.

### 4.4. Activin-A and Obesity

Immunoinflammatory cells that accumulate within the adipose tissue of individuals with obesity may lead to fat mass enlargement through the paracrine effects on progenitor cells. In obesity, the growth of white adipose tissue results from an increase in the size and number of adipocytes. Knowing that mature adipocytes do not divide, new adipocytes are formed by adipose progenitors [51]. In other words, the differentiation potential of human preadipocytes is inversely correlated with obesity, whereas precursor cells are positively correlated to BMI [52,53]. Activin-A seems to be involved in this process; in humans, it is secreted by undifferentiated adipocyte progenitors and, reversely, it increases the number of undifferentiated adipose-derived stem cells [54]. In line with the above is our finding of significantly increased serum levels of activin-A in children and adolescents with obesity.

In addition, activin-A levels have been associated with metabolic syndrome in adults [55], with cardiometabolic disturbances that may lead to heart failure, myocardial fibrosis via activation of the ERK1/2 and p38-MAPK pathways [56], and myocardial infarction. Therefore, the activin-A pathway has been proposed as a potential therapeutic target for obesity-associated cardiometabolic complications. Whether our finding of increased levels of activin-A in children and adolescents with obesity suggests increased risk for future cardiometabolic morbidity requires further exploration.

### 4.5. Activin-A and Type-2 Diabetes

Activin-A has been correlated with parameters of insulin resistance, such as homeostasis model assessment of insulin resistance (HOMA-IR) and fasting plasma insulin concentrations [57]. Also, in patients with prediabetes and normal glucose levels, an association between cIMT, a marker of carotid and coronary atherosclerosis, and activin-A levels has been established [57].

### 4.6. Activin-A and Type-1 Diabetes

As pertains to T1DM, activin-A is expressed in islet cells, which suggests that it has a role in glucose homeostasis. Also, *INHBA*, the gene encoding activin-A, is the most prominently expressed member of the TGFβ superfamily in cultured, functional human islets [58]. However, thus far there have been no literature data on the possible role of activin-A in T1DM, either in adult or in pediatric populations. Our findings show that a duration of T1DM ≥ 5 years is positively correlated with activin-A concentrations, which may indicate a possible role as the disease progresses. However, the findings of the current study failed to demonstrate a significant difference in activin-A serum concentrations between children and adolescents with T1DM and the control group. This finding may be due to the young age and the short duration of diabetes in most of the participants. It could be hypothesized that in larger and older populations with T1DM, and with longer durations of diabetes, statistical significance may have been reached.

### 4.7. Clusterin and Obesity

Circulating plasma clusterin concentrations have been associated with inflammation and increased cardiovascular risks in patients with obesity, however, the underlying mechanism remains elusive [59,60]. The expression of the CLU gene is increased in patients with obesity compared to lean subjects, and is decreased after bariatric surgery and subsequent weight loss [61].

The present study showed no statistically significant difference in the serum levels of clusterin between children and adolescents with obesity. This may be due to the young age and the absence of metabolic disorders in the participating children and adolescents, but additional studies are needed so that the role of clusterin in pediatric obesity can be clarified.

### 4.8. Clusterin and Type-2 Diabetes

Urinary clusterin levels have been found to be significantly increased in patients with T2DM compared to control groups, and this has been associated with the annual decline in the estimated glomerular filtration rate (eGFR) and the progression of DN stage [62]. Our finding of a negative correlation between clusterin and eGFR agrees with these data.

In addition, glomerular clusterin levels were found to be higher in 10 patients with diabetes compared to the control group [63]. Clusterin is also upregulated in the glomeruli of adult patients with kidney disease, including diabetic nephropathy, and it is believed to have a protective role against oxidative stress and to protect podocytes against oxidative stress-induced apoptosis [64,65]. Finally, there are studies to suggest that clusterin may be a protective factor against kidney fibrosis [66].

Our findings showed no significant difference in serum clusterin levels between participants with T1DM and the control group, which may be due to the short mean duration of diabetes without evidence of kidney damage. However, it is not clear whether glomerular clusterin is synthesized by glomerular cells or derived from the circulation [62]; therefore, serum clusterin concentrations may not be representative of glomerular clusterin. Future research may help elucidate the role of clusterin in T1DM as well.

## 5. Limitations and Strengths

The present study has some limitations. It is a retrospective case-control study, and therefore a cause–effect relationship between the studied markers and obesity or T1DM and their complications cannot be confirmed. Longitudinal studies would be more appropriate for drawing safer conclusions. Moreover, the sample used was relatively small, although not smaller than that of many other similar studies in adults. Another limitation is that serum concentrations are not necessarily representative of the adipose or other tissue concentrations of the studied markers.

Among the strengths of this study is that it provides additional data on the association between MCP-1 and pediatric obesity. Additionally, to our knowledge, it is the first study to investigate the possible involvement of activin A and clusterin in the pathophysiology of pediatric obesity, and the potential role of MCP-1, activin-A, and clusterin in T1DM. Our data contribute to the ongoing research on the pathological processes implicated in the development of these two chronic diseases and their complications. Chemokines produced by monocytes, such as MCP-1, and proteins produced in the adipose tissue of pediatric populations with obesity, such as activin-A, may reflect the inflammatory process in the adipose tissue, since adipose tissue collection for pediatric populations is unethical. Similarly, if the relationship between circulating concentrations of proteins that are also present in the pancreatic islets is confirmed, useful diagnostic or predictive biomarkers will have been discovered. Of course, extensive future research is required to confirm these hypotheses.

## 6. Conclusions

Our finding that inflammation markers such as MCP-1, activin-A, and clusterin are not altered in children with T1DM is of particular interest. In contrast, obesity in children is positively correlated to serum MCP-1 values and characterized by higher activin-A levels, which may reflect an already established systematic inflammation in children with obesity from a young age. This may also be an indication that obesity represents a bigger health burden compared to T1DM for young people.

The biomarkers measured in this present study reflect adipose tissue biology and macrophage-mediated inflammation, providing further insight into the pathophysiology of obesity and T1DM in children and adolescents. The enhancement of the understanding of the underlying pathophysiological processes involved in these two medical conditions may prove useful for the diagnosis, monitoring, or prediction of future complications, as well as for the development of potential treatment approaches. These areas await further research.

## Figures and Tables

**Figure 1 diagnostics-14-00450-f001:**
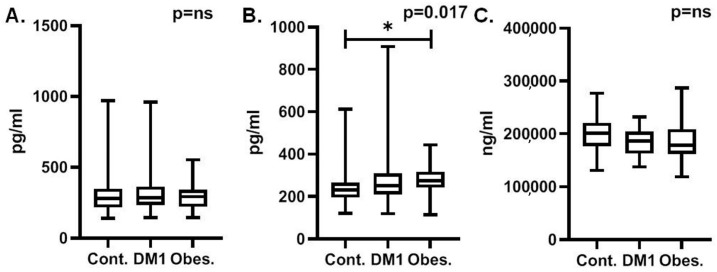
Comparison of MCP-1 (**A**), activin-A (**B**), and clusterin (**C**) serum levels between control group participants (contr.), patients with type-1 diabetes (DM1), and patients with obesity (Obes.). (*: *p* < 0.05, ns: non-significant).

**Table 1 diagnostics-14-00450-t001:** Basic clinical and biochemical characteristics of control participants and patients with T1DM and obesity (ns: non-significant).

	Controls (*n* = 43)	T1DM (*n* = 43)	Obesity (*n* = 43)	*p*-Value (ANOVA for Quantitative Variables)
Age (years)	12.3 ± 4.3	12.6 ± 3.8	12.4 ± 3.8	ns
Gender (males/females)	12/31	20/23	18/25	ns
BMI z-score	0.61 ± 0.73	0.59 ± 0.89	2.17 ± 0.45	<0.001
Serum creatinine (mg/dL)	0.66 ± 0.13	0.7 ± 0.12	0.65 ± 0.13	ns
eGFR (mL/min/1.73 m^2^)	95.4 ± 17.3	92.6 ± 21.5	97.6 ± 15.4	ns
TSH (mIU/L)	2 ± 0.86	1.9 ± 0.79	2.45 ± 1.1	0.015
FT4 (ng/dL)	1.27 ± 0.18	1.25 ± 0.17	1.24 ± 0.21	ns
Cholesterol (mg/dL)	161 ± 26.4	167 ± 29.7	163 ± 37.8	ns
LDL (mg/dL)	93 ± 28.7	94 ± 23.9	95 ± 36.3	0.04
HDL (mg/dL)	61 ± 13.1	60 ± 11.2	50 ± 12.7	<0.001
Triglycerides (mg/dL)	63 ± 21.6	66 ± 26.8	89 ± 56.4	0.003

**Table 2 diagnostics-14-00450-t002:** Monocyte chemoattractant protein-1, activin-A, and clusterin serum between control participants and patients with either T1DM or obesity (ns: non-significant).

	Controls	T1DM	Obesity	*p*-Value (ANOVA for Clusterin and Kruskal–Wallis Test for MCP-1 and Activin-A)
MCP-1 (pg/mL)	305.5 ±143.4	326.8 ± 164.2	297.6 ± 99.35	ns
Activin-A (pg/mL)	244.5 ± 89.85	278.1 ± 125.4	278.2 ± 68.27	0.0168
Clusterin (ng/mL)	200,023 ± 34,656	186,222 ± 25,739	186,679 ± 39,289	ns

## Data Availability

The data presented in this study are available on request from the corresponding author. The data are not publicly available due to privacy.

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
