# Peer review of "Monocyte Chemoattractant Protein-1 (MCP-1), Activin-A and Clusterin in Children and Adolescents with Obesity or Type-1 Diabetes Mellitus"

_diagnostics, 2024, doi:10.3390/diagnostics14040450_

Round 1
Reviewer 1 Report
Comments and Suggestions for Authors
In the present submission, the authors reported that the alterations of inflammation markers MCP-1, activin-A and clusterin in children with T1DM and obesity. A few concerns about the submission.
(1) The conclusion should be optimized. The current one is obscure.
(2) In table 1. the authors are strongly suggested to check the original data about the BMI, TSH and others, make sure the statistical method is correct.
(3) In Table 2 and Figure 1B, make sure there is a difference in MCP-1 between groups. It seems there is no significant difference between groups.
(4)The correlation between MCP-1 and serum lipid markers should be conducted.
Author Response
Point by point reply to comments of Reviewer 1
Dear Reviewer,
Thank you for reviewing our manuscript entitled “Chemoattractant Protein-1 (MCP-1), Activin-A and Clusterin in Children and Adolescents with Obesity or Type-1 Diabetes Mellitus” (Diagnostics-2791101). We appreciate the time and efforts that you dedicated to provide valuable comments on our article.
We have now completed the revisions required. All changes made are highlighted in yellow in the revised manuscript. A point-by point answer to the comments made by the reviewers is provided below.
The revised manuscript has been viewed and approved by all authors.
We hope that our revised manuscript will meet your requirements for publication in Diagnostics.
Sincerely,
Xenophon Sinopidis, MD, PhD
Associate Professor of Pediatric Surgery
Comments to author
In the present submission, the authors reported the alterations of inflammation markers MCP-1, activin-A and clusterin in children with T1DM and obesity. A few concerns about the submission.
Comment 1.
The conclusion should be optimized. The current one is obscure.
Reply:
The conclusion has now been optimized. We believe that in its revised form it clearly conveys the message that the biomarkers measured in the study reflect adipose tissue biology and macrophage-mediated inflammation, providing further insight into the pathophysiology of obesity and T1DM in children and adolescents.
Comment 2.
In table 1 the authors are strongly suggested to check the original data about the BMI, TSH and others, make sure the statistical method is correct.
Reply:
All quantitative variables in Table 1 were examined with ANOVA for testing the difference of means between controls and patients with either T1DM or obesity. The p value presented in table is referring to the ANOVA summary result and no post-hoc tests were evaluated. All tests were double checked, and no inconsistencies were found. This is now indicated to the revised Tables 1 and 2.
Comment 3.
In Table 2 and Figure 1B, make sure there is a difference in MCP-1 between groups. It seems there is no significant difference between groups.
Reply:
Indeed, there is no statistical difference in serum MCP-1 values between children and adolescents with either type 1 diabetes or obesity and controls. This is stated both in the ‘Results' section and shown in Table 1 (p value: non-significant) and in Figure 1A (not B). In Figure 1B, a statistically significant difference in serum activin-A levels is shown between children and adolescents with obesity and controls.
Comment 4.
The correlation between MCP-1 and serum lipid markers should be conducted.
Reply: We performed bivariate Spearman's correlation test to examine the association between serum total cholesterol, LDL, HDL and triglycerides levels with serum MCP-1. We found no significant correlations between total cholesterol, LDL, HDL or triglycerides and MCP-1 serum levels when examined either in the total population of the cohort or separately in patients with T1DM or obesity. This is now mentioned in the ''Results'' section.
Reviewer 2 Report
Comments and Suggestions for Authors
The authors present an interesting investigation of type 1 diabetes never less the authors declare in the aims of the study the "possible value" of MCP-1, etc., in Type 1 DM in children and adolescents, which is inconsistent with the title statement of Diagnostic and Prognostic...
Please reconsider the title change, adjusting to the possibility pending the results and the applicability of these markers to a wide population.
Please enrich the introduction in line 66 with an explanation of the role of MCP-1 in T2DM and the correlation with T1DM.
Please consider the separation of the biomarkers used in the study as a sub-index, as an example for type 1 diabetes and obesity, and in the conclusions, make a correlation statement if it exists.
Please insert the methods of ethics approval and when samples were collected and analyzed for this study.
The authors did not declare other excluding criteria in line 88 or the origin of control group samples.
Authors need to explain widely the statement between lines 227 to 232 since the MCP-1 analysis declares in the study is for the underage population, and they assume the partially supported correlation of MCP-1 in adults without a properly referenced comparison.
Comments on the Quality of English LanguageEnglish editing service is required
Author Response
Point by point reply to comments of Reviewer 2
Dear Reviewer,
Thank you for reviewing our manuscript entitled “Chemoattractant Protein-1 (MCP-1), Activin-A and Clusterin in Children and Adolescents with Obesity or Type-1 Diabetes Mellitus” (Diagnostics-2791101). We appreciate the time and efforts that you dedicated to provide valuable comments on our article.
We have now completed the revisions required. All changes made are highlighted in yellow in the revised manuscript. A point-by point answer to the comments made by the reviewers is provided below.
The revised manuscript has been viewed and approved by all authors.
We hope that our revised manuscript will meet your requirements for publication in Diagnostics.
Sincerely,
Xenophon Sinopidis, MD, PhD
Associate Professor of Pediatric Surgery
Comments to author
The authors present an interesting investigation of type 1 diabetes never less the authors declare in the aims of the study the "possible value" of MCP-1, etc., in Type 1 DM in children and adolescents, which is inconsistent with the title statement of Diagnostic and Prognostic.
Reply: We have now changed the title so that there is no inconsistency with what is declared in the Aims of the study.
Comment 1.
Please reconsider the title change, adjusting to the possibility pending the results and the applicability of these markers to a wide population.
Reply: We agree with the reviewer that the cross-sectional design of our study is not appropriate for the establishment of the prognostic value of the examined serum markers. The prognostic value should be examined in a wider population with prospective long follow up. Thus, we have altered our title as follows: ''Chemoattractant Protein-1 (MCP-1), Activin-A and Clusterin in Children and Adolescents with Obesity or Type-1 Diabetes Mellitus''.
Comment 2.
Please enrich the introduction in line 66 with an explanation of the role of MCP-1 in T2DM and the correlation with T1DM.
Reply:
We have added to the Introduction a brief explanation of the role of MCP-1 in T2DM and T1DM (lines 66-69). More information is provided in the Discussion section, and specifically in the sub-section “MCP-1 and Diabetes”.
Comment 3.
Please consider the separation of the biomarkers used in the study as a sub-index, as an example for type 1 diabetes and obesity, and in the conclusions, make a correlation statement if it exists.
Reply:
In the Discussion section, we have previously used separate sub-indexes for each marker and obesity or diabetes. In the revised version of the manuscript, we have further separated the studied biomarkers using a sub-index for each biomarker and obesity, type 2 diabetes or type 1 diabetes. We hope that we understood the reviewer’s comment correctly.
Comment 4.
Please insert the methods of ethics approval and when samples were collected and analyzed for this study.
Reply:
All samples were obtained starting on January 2021 until December 2022 and were measured between May and July 2023. The study was approved by the Research Ethics Committee of the University Hospital of Patras (number of approval: 353/02/09/2015) and is in accordance with the Helsinki Declaration as revised in 2013. Written informed consent was obtained from the parents or legal representatives of the participating children and adolescents and informed assent was obtained from the participants. This is clearly now stated in ‘Methods'.
Comment 5.
The authors did not declare other excluding criteria in line 88 or the origin of control group samples.
Reply:
Thank you for this particular comment. All patients and controls included in the study were recruited during a standard morning visit to the outpatient clinic of the Pediatric Endocrinology Department of the University Hospital of Patras. Patients or controls with a history of acute febrile illness during the previous 2 weeks, other endocrinopathies (hypothyroidism, adrenal insufficiency etc.) or chronic diseases (cancer, asthma and malabsorption) were excluded from the study. As controls we used children and adolescents that were externally referred to the outpatient clinic and subsequently proved that had no substantial medical issue. This is now added in ‘Methods' in the revised version of our manuscript.
Comment 6.
Authors need to explain widely the statement between lines 227 to 232 since the MCP-1 analysis declared in the study is for the underage population, and they assume the partially supported correlation of MCP-1 in adults without a properly referenced comparison.
Reply:
We thank the reviewer for the comment. We have now added literature data supporting similar findings in the adult and pediatric population (lines 233-240).
Round 2
Reviewer 1 Report
Comments and Suggestions for Authors
The concerns have been addressed.
Reviewer 2 Report
Comments and Suggestions for Authors
The authors present a revised version of the original manuscript, the modifications are consistent with the observations.
Comments on the Quality of English LanguageMinor text editing is required.